# OpenReview forum: "Hierarchical Representations for Cross-task Automated Heuristic Design using LLMs"
_ICML.cc/2026/Conference — ICML 2026 regular_

### Official Review · Reviewer_Bg7V · 2026-02-28

**Soundness:** 2
**Presentation:** 3
**Significance:** 2
**Originality:** 2
**Overall Recommendation:** 3
**Confidence:** 4

**Summary:**

This paper introduces a novel automatic heuristic design framework called MTHS, which aims to break through the efficiency bottleneck of traditional algorithm design by leveraging large language models. This method adopts a multi-task learning perspective and combines a general high-level meta-heuristic strategy with specific low-level task programs by maintaining a hierarchical evolving population. Its core innovation lies in the introduction of a cross-task knowledge transfer mechanism, enabling the model to apply the successful patterns discovered in a certain optimization problem to other related fields. The experimental results show that MTHS has reached the top level in benchmark tests such as the Traveling Salesman Problem and Workshop Job Scheduling, and even refreshed some of the optimal solutions. Compared with the single-task method, this framework significantly reduces the number of code evaluations and Token consumption, greatly enhancing the economic efficiency of automated algorithm discovery. This hierarchical representation method not only enhances the generalization ability of the algorithm, but also successfully transfers the design experience of combinatorial optimization to black-box optimization tasks.

**Compliance With Llm Reviewing Policy:**

Affirmed.

**Key Questions For Authors:**

1) During the process of multi-task evolution, have you ever observed the phenomenon of "negative migration" (that is, the addition of one task actually reduces the performance of other tasks)? How does the framework currently detect and mitigate this kind of interference?

2) How can computing efficiency be improved to enhance its usability in the real world?

3) As the number of tasks m increases, Pareto Dominance is prone to the "curse of dimensionality", resulting in the majority of individuals in the group not dominating each other. If the task expands to more than 10, can the current population selection strategy still effectively distinguish the superiority and inferiority of the algorithm? How should the parameters of the population be set properly? Does an overly large population affect search efficiency?

4) Have you found that some models are stronger in the design of meta-heuristic logic (at a higher level), while others have an advantage in the refinement of code implementation (at a lower level)?

**Limitations:**

yes

**Strengths And Weaknesses:**

**Strengths**
1) Unlike the existing AHD methods that can only design a Monolithic algorithm for a specific task, MTHS adopts a hierarchical representation, separating the general meta-heuristic logic (MH) from the program implementation of a specific task.

2) When handling multiple tasks simultaneously, MTHS is more efficient than running multiple single-task evolutions independently. In the experiment handling three tasks, it reduced the number of code evaluations by approximately 67% and lowered the overall Token consumption.

3) Explicit cross-task Knowledge Transfer has been introduced, allowing excellent algorithmic patterns discovered in one task to be extracted and applied to other related tasks, thereby accelerating the design process and enhancing algorithm robustness.

4) Research shows that compared with pure Thought or Pseudocode representations, the template-based hierarchical representation used by MTHS can significantly improve search efficiency and enable the model to convergence more quickly in complex search Spaces.

**Weakness**
1) Although it is more efficient than the single-task method, a single multi-task evolutionary run still consumes a large amount of computing resources (for example, running three tasks on a workstation equipped with dual cpus takes about 1.5 days).

2) Although MTHS is robust to different models, the quality of the heuristic algorithms designed is still limited by the capabilities of the underlying LLM.

3) It is still necessary to conduct in-depth analysis on when and why knowledge transfer can succeed, as well as how to directly incorporate resource and reliability constraints into the search process.

4) A great deal of effort was spent comparing many old algorithms.

---

> ### Author Rebuttal · Authors · 2026-03-31
>
> Thank you very much for your time and effort in reviewing our work. We address your concerns point-by-point as follows:
>
> > **W1 & Q2** Computing resources and how to improve?
>
> We agree that the efficiency could be further improved; but we do not view computational cost as a key limitation for LLM-driven AHD. Our position is grounded in three concrete facts:
>
> + **Relative efficiency**: Our framework is demonstrably more efficient than existing SOTA LLM-driven AHD methods (see **Table 8, Appendix**).
> + **Absolute cost:** A complete search requires only about one day, which is much faster than traditional automated heuristic design approaches such as deep learning–based solvers, which typically need up to one week of training.
> + **Reusability**: Once the metaheuristic is designed, it can be reused across many related tasks, as demonstrated in our generalization study, providing strong amortized efficiency.
>
> We also identify four actionable strategies to further reduce resource usage: **(i)** incorporate lightweight screening evaluations before full trials, **(ii)** cache and reuse promising task-specific components, **(iii)** warm-start the search using previously discovered heuristics, and **(iv)** more aggressively parallelize low-level evaluations. These will be added to our revised discussion section as concrete future work.
>
> > **W2:** The quality of the heuristic algorithms designed is still limited by the capabilities of the underlying LLM.
>
> We appreciate the insight. Indeed, the absolute performance of any LLM-driven AHD method, including our baselines (EoH and ReEvo), depends on the reasoning capability of the underlying model. The methodological contribution of our framework is orthogonal to the specific choice of the underlying LLM. Improvements in model capability can enhance absolute performance, but do not diminish the value of our algorithmic design.
>
> Empirically, **Appendix E6–E7** shows that MTHS consistently improves performance across different LLM families, including open-source (Qwen, DeepSeek) and smaller proprietary models (GPT‑3.5). This demonstrates that our method generalizes across model capabilities. We will clarify this point in the revision.
>
> > **W3 & Q1** In-depth analysis on knowledge transfer, negative migration, and incorporate resource and reliability constraints into the search process.
>
> Yes, negative transfer is possible in principle. The current framework mitigates it in two ways. **First**, transferred programs are only accepted if they improve the target-task score; otherwise, they are discarded. **Second**, the high-level population is managed by task-wise elitism plus Pareto ranking, which prevents one task from dominating all others and preserves per-task champions.
>
> Empirically, **Table 17** indicates that transfer is beneficial overall on TSP and CVRP, while the ASP results in **Table 5** illustrate that transfer is *not* universally beneficial across all task pairs. We will clarify that the current mechanism is a conservative accept-if-better filter rather than a full negative-transfer detector.
>
> We agree that integrating resource awareness directly into the search is a meaningful next step. Future versions will treat time, token cost, and failure rate as explicit objectives or constraints during survival and mutation.
>
> > **W4:** Many old algorithms.
>
> We clarify that our baselines were chosen to cover representative metaheuristics as well as the most relevant recent LLM-driven AHD methods, ensuring breadth and relevance.
>
> + Many of them are SOTA methods, e.g., LKH and HGS are widely recognized SOTA metaheuristics for TSP and CVRP, respectively.
>
> + Moreover, the LLM-driven baselines EoH, ReEvo, and MCTS-AHD are STOA methods and were published recently on ICML and NeurIPS in 2024, 2024, and 2025, respectively.
>
>
> > **Q3:** The number of tasks and population size.
>
> We have conducted additional experiments involving four tasks (see **Reviewer gkWt Q1**).  For scenarios involving more tasks (e.g., >10), many-objective techniques such as decomposition/reference-vector methods, task clustering, or hybrid scalarization can be more suitable. Population size should increase with the number of tasks only moderately, since each additional individual is expensive to evaluate; making it too large would indeed reduce efficiency.
>
> > **Q4:** Different models are good at different levels?
>
> This is an interesting possibility, but we have not yet run a controlled study that cleanly separates high-level metaheuristic generation from low-level code refinement across models. Our current results mainly show that stronger models tend to perform better overall. We agree that a cross-model hybrid setting (one model for high-level search, another for low-level refinement) is promising, and we will mention this as an important future direction.

---

> > ### Author Rebuttal · Reviewer_Bg7V · 2026-04-03
> >
> > Thank you to the author for the tremendous efforts made in the reply, especially for the detailed explanations and supplementary data provided regarding computational efficiency and negative migration mechanisms. The author successfully clarified the logical consistency of the MTHS framework in multi-task scenarios. However, although these technical details have been refined, I still believe that this work meets my initial judgment in terms of core innovation (incremental contribution compared to the existing EoH or ReEvo frameworks). Therefore, I have decided to maintain the original score and suggest that the author incorporate the discussion on "resource awareness" mentioned in the reply into the revised edition.
> >
> > **Updated information**:
> >
> > I appreciate the effort put into the rebuttal; however, my core concerns remain unresolved for the following reasons:
> >
> > 1.  **Imbalance in Baseline Comparisons**:
> >     *   The manuscript devotes significant space to comparing the proposed method against **classic heuristics from the 1970s** (e.g., NN, Insert, and GUPTA). While these are necessary for completeness, their relevance in a top-tier machine learning conference like ICML is limited when discussing the cutting edge of **LLM-driven Automated Heuristic Design (AHD)**.
> >     *   The primary value of this work should be measured against **2024/2025 SOTAs like EoH and ReEvo**. In my assessment, the hierarchical representation—while a structured architectural choice—serves more as an **engineering optimization** of the "thought-and-code" paradigm rather than a fundamental breakthrough in search logic.
> >
> > 2.  **Lack of New Empirical Evidence in Rebuttal**:
> >     *   Regarding my technical questions on **"negative transfer"** and **"multi-task scalability"**, the rebuttal largely **restated the logic already present in the manuscript** .
> >     *   I expected the rebuttal to provide **new targeted experimental data**—such as performance impact curves when tasks are increasingly dissimilar, or specific Pareto front transitions with more than 10 tasks—to demonstrate robustness. Repeating the existing methodology does not provide the empirical proof needed to resolve these technical doubts.
> >
> > 3.  **Generalization Boundaries**:
> >     *   The framework's failure to generalize to the **Admissible Set Problem (ASP)** suggests that the current "knowledge transfer" is highly dependent on the LLM's pre-existing optimization priors for related combinatorial tasks, rather than the discovery of a truly domain-agnostic meta-logic.
> >
> > While the authors have clarified certain technical details, the **overall contribution remains incremental** in the context of the rapidly evolving AHD field. The work represents a solid combination of existing evolutionary ideas applied to a multi-task setting, but it does not yet demonstrate a paradigm shift that warrants a higher score.

---

> > > ### Author Response · Authors · 2026-04-03
> > >
> > > Thank you again for your careful reading of our paper and for acknowledging that our technical concerns have been clarified, especially regarding computational efficiency, negative transfer mitigation, and the logical consistency of MTHS in the multi-task setting. We sincerely appreciate your time and the constructive engagement.
> > >
> > > We would like to respectfully respond to your remaining point about **“core innovation” / “incremental contribution.”**
> > >
> > > Our understanding is that your original review focused on several concrete technical concerns: computational cost, transfer behavior (including negative migration), scalability with multiple tasks, and the role of different LLMs. In our rebuttal, **we directly addressed each of these points with additional explanations, ablations, and new experimental evidence. We are grateful that you recognized these concerns as clarified**.
> > >
> > > However, the remaining reason for maintaining the score now appears to be a concern about “lack of innovation” or the contribution being “incremental” relative to EoH/ReEvo. **We respectfully note that this issue was not raised as a primary weakness in the original review, nor was it one of the specific technical questions posed for rebuttal**. By contrast, **the originality of our work is precisely the main aspect that distinguishes it from prior LLM-based AHD methods**:
> > >
> > > + Hierarchical representation: Unlike EoH/ReEvo, which evolve monolithic task-specific heuristics, our method explicitly separates task-agnostic metaheuristic logic from task-specific executable programs.
> > >
> > > + Cross-task co-evolution: MTHS does not optimize each task independently. It performs joint multi-task search, enabling the discovery of reusable search principles across problems.
> > >
> > > + Explicit knowledge transfer across tasks: We introduce a mechanism that transfers promising low-level designs between tasks under an accept-if-better rule, which is absent in prior LLM-based AHD frameworks.
> > >
> > > + Generalization as a central goal, not a side effect: Existing methods mainly target single-task heuristic discovery. In contrast, our paper studies cross-task automated heuristic design and evaluates transfer to unseen problems, including both combinatorial optimization and black-box optimization.
> > >
> > > In fact, these strengths were also reflected in **your original review (in particular, strengths 1, 3, and 4)**. You also described our work as a **“novel automatic heuristic design framework ... break through the efficiency bottleneck of traditional algorithm design by leveraging large language models,”** and further noted that its **“core innovation lies in the introduction of a cross-task knowledge transfer mechanism.”**. For this reason, we believe the paper’s contribution is more than an incremental extension of EoH or ReEvo. **It changes the representation, the search space, the optimization setting, and the intended outcome** of LLM-based AHD.
> > >
> > > Of course, we fully agree that the revised paper should make this novelty clearer. We will strengthen the final version accordingly, including a more explicit comparison to EoH/ReEvo and a clearer discussion of resource awareness, as you suggested.
> > >
> > > With that said, since your original technical concerns have now been addressed, we **respectfully ask that the paper be judged primarily on the resolved review criteria and the demonstrated contribution in the submitted work, rather than on a new concern that was not part of the initial review basis**. We would be very grateful if you could reconsider whether the current score still best reflects your own updated assessment of the manuscript.
> > >
> > > Thank you again for your valuable feedback and thoughtful consideration.

---

### Official Review · Reviewer_gkWt · 2026-03-05

**Soundness:** 3
**Presentation:** 3
**Significance:** 3
**Originality:** 3
**Overall Recommendation:** 4
**Confidence:** 3

**Summary:**

This paper proposes a cross-task heuristic generation framework driven by LLMs, MTHS, featuring a two-level evolutionary mechanism. The high-level component evolves task-agnostic metaheuristics that encapsulate general problem-solving logic, while the low-level component instantiates these metaheuristics into task-specific executable programs tailored to individual optimization tasks. The authors evaluate on four COPs, and further generalize the heuristics on the unseen tasks. Compared to existing LLM-based methods that directly evolve task-specific heuristics, MTHS achieves superior performance.

**Compliance With Llm Reviewing Policy:**

Affirmed.

**Final Justification:**

The authors have addressed my concerns and the method demonstrates impressive performance.

**Key Questions For Authors:**

1. Could you please give more results with different training problems and analyze the generalization performance with different training problems.
2. Please analyze the metaheuristics found by MTHS and the failed cases. In addition, the authors should analyze the complexity of metaheuristics found by MTHS compared to the classic algorithms.

**Limitations:**

yes

**Strengths And Weaknesses:**

Generalization has long been a persistent challenge for traditional heuristic algorithms. While this paper advances the field by designing metaheuristics via LLMs, I still remain some concerns.
1. COPs are problem-specific by nature, even tiny changes to the objective function or constraints can break existing algorithms. The key to generalization lies in extracting transferable patterns between problems. But the TSP and CVRP used in this paper are quite similar. I am curious: can the framework still find good metaheuristics when dealing with vastly different problems?
2. How does the paper balance the trade-off between task-agnostic generality and task-specific adaptability? If the training problems are too diverse, will the resulting heuristics become overly general, just like Simulated Annealing? And how exactly do the training problems impact the final evolution results?
3. The paper lacks interpretability for the generated metaheuristics. The authors need to better explain how the knowledge transfer actually works. Does it hold up for similar problems, or does it break down when the problem structure changes significantly?

---

> ### Author Rebuttal · Authors · 2026-03-31
>
> Thank you very much for your time and effort in reviewing our work. We address your concerns point-by-point as follows:
>
> > **W1:** Can the framework still find good metaheuristics when dealing with vastly different problems?
>
> + **Within the training phase**, we did not rely only on TSP/CVRP; we also included FSSP, which differs substantially in representation and constraints. We have also added an additional experiment with 4 training problems in the response to **Q1**.
>
> + **In the generalization phase**, we further tested the discovered metaheuristic on BPP, BBOP, and ASP. The results show that MTHS generalizes well to BPP and BBOP, but not uniformly to ASP. For example, on BPP, the MTHS-guided Gemini-2.5-pro solver reaches 0.34% and 0.25% gap on the two test sets, outperforming MCTS-AHD at 0.48% and 0.53% (**Table 3**). On BBOP, GPT-5-mini + MH reduces the average score by almost $9\times$ (2.86 to 0.32, **Table 4**). We view this as evidence that the method **can transfer beyond very similar routing tasks, while also confirming that generalization has clear limits when the mismatch becomes too large.
>
> > **W2:** How does the paper balance the trade-off between task-agnostic generality and task-specific adaptability?
>
> The balance is handled by the hierarchical design itself. The high level evolves a task-agnostic scaffold, while the low level instantiates and refines task-specific programs. This prevents the final solver from collapsing into an overly generic method such as a bare simulated annealing loop. In other words, generality is enforced at the level of search *logic*, while adaptability is preserved at the level of implementation. We agree that this trade-off should be discussed more explicitly, and we will expand the paper to explain that diverse-but-related training tasks are beneficial, whereas excessively heterogeneous tasks may weaken transfer and require clustering or more structured task grouping.
>
> > **W3 & Q2** The paper lacks interpretability for the generated metaheuristics? Please analyze the metaheuristics found by MTHS and the failed cases. In addition, the authors should analyze the complexity of metaheuristics found by MTHS compared to the classic algorithms.
>
> **On the success side**, the discovered metaheuristics tend to combine memory, adaptive operator choice, constructive seeding, and repair. Because effective BBOP algorithms also rely on iterative search, these generic optimization principles are domain-agnostic and thus remain valuable. **On the failure side**, the generalization experiments show that transfer is not universal: it is strong on BPP, still useful on BBOP, and weak on ASP. The ASP demands a constructive algorithm; therefore, the iterative search ideas may not be helpful.
>
> **Qualitatively**, the metaheuristics discovered by MTHS exhibit a computational structure similar to hybrid population--memory--local-search frameworks, where the main time consumption still lies in the neighborhood evaluation and local repair phases. Let $n$ denote the number of decision variables and $N_{\mathrm{eval}}$ the number of fitness evaluations. Then the overall complexity of these learned algorithms can be approximated as $\mathcal{O}\big(N_{\mathrm{eval}} \cdot C_{\mathrm{neigh}}\big),$
>
> where $C_{\mathrm{neigh}}$ is the cost of computing local modifications and constraint repairs---identical in order to that of classical memetic or tabu-based search.
>
> The additional adaptive and cooperative mechanisms (e.g., adaptive operator selection, shared memory update, or probabilistic perturbation choice) introduce only linear or sublinear overhead $\mathcal{O}(n)$ per iteration, which is negligible relative to the local search and evaluation terms. Therefore, compared with classical Memetic Algorithms, Tabu Search, or Simulated Annealing, the MTHS-discovered heuristics operate with **comparable theoretical complexity** but potentially **different constant factors** reflecting richer state management. We will add a detailed discussion on complexity to the revised version.
>
> > **Q1:** Could you please give more results with different training problems and analyze the generalization performance with different training problems.
>
> We have conducted additional results trained on four different problems (TSP, CVRP, FSSP, and BBOP).
>
> While the results on TSP and CVRP are slightly sacrificed, the results on BBOP are significantly improved, which means that MTHS can still design general metaheuristics when more problems are used and provide good trade-offs on all training problems.
>
> |  | TSP | CVRP | FSSP | BBOP |
> | --- | --- | --- | --- | --- |
> | MTHS on 3 problems | 0.005646 | 0.0412245 | 0.141799 | 0.02880 (test) |
> | MTHS on 4 problems | 0.00994 | 0.04487 | 0.14147 | 0.00003 |

---

> > ### Author Rebuttal · Reviewer_gkWt · 2026-04-02
> >
> > Thanks for your rebuttal. However, I still have some concerns:
> > 1. Regarding W1 & Q1, I remain skeptical about the generalization capability of such meta-heuristics. Given the intricate structures of COPs, the authors should consider evaluating on more diverse structures during training or testing, such as MaxCut or MIS. Furthermore, even for the same constraint structure, heuristics often change when scaling from small to very large instances. Therefore, the claimed generalization ability, especially for heuristics discovered by LLMs, is not entirely convincing. Additionally, it appears that performance degrades when the training problem structure differs significantly from the testing one.
> > 2. As for the meta-heuristics, when generalizing to different problems, is the improvement brought by them truly significant? Or does it incur substantial overhead at the low-level? I would like to understand the underlying mechanism of these meta-heuristics.
> >
> > So, I maintain my score.

---

> > > ### Author Response · Authors · 2026-04-02
> > >
> > > Thank you for your reply. We respond point-by-point below and would greatly appreciate it if you could reconsider your assessment.
> > >
> > > > On your concern, “skeptical about the generalization capability of such meta-heuristics.”
> > >
> > > We have now conducted additional experiments on **MaxCut** and **MIS**. Specifically, we conducted additional experiments with a training set including **five tasks**: **MaxCut, MIS, FSSP, TSP, and CVRP**, using the same settings and budget as in the original setup except for adding the two new tasks. This substantially increases the structural diversity of the training problems.
> > >
> > > We then evaluate the metaheuristic designed by MTHS on **MaxCut** and **MIS**, and compare it with representative metaheuristics including **GA**, **ILS**, and **Tabu Search**. The testing results on **MaxCut**, across different distributions and sizes, are: **GA: 5387.4, ILS: 5602.9, Tabu: 5605.0, and MTHS: 5611.8**. The testing results on **MIS**, across different distributions and sizes, are: **GA: 29.4, ILS: 28.7, Tabu: 27.3, and MTHS: 29.6**. These results show that MTHS achieves the best average performance on both tasks, even when trained on a much more diverse set of combinatorial optimization problems. Detailed results on different distributions are available at: https://anonymous.4open.science/r/ICML2026-F1D9.
> > >
> > > More broadly, we have already evaluated on **8 different tasks**. We not only demonstrated good and efficient generalization to most tasks, but also identified limitations on ASP. We believe these results provide strong evidence for both the effectiveness and the boundaries of generalization.
> > >
> > > > On your concern, “even for the same constraint structure, heuristics often change when scaling from small to very large instances. Therefore, the claimed generalization ability, especially for heuristics discovered by LLMs, is not entirely convincing.”
> > >
> > > We acknowledge the cross-distribution generalization challenge within a single task (or constraint structure). Indeed, this is one of the key research topics in LLM-driven AHD [1][2]. However, we study a different topic—**cross-problem metaheuristic design with LLMs**—which is, to our knowledge, the first work of its kind on cross-problem study for LLM-driven AHD (as also mentioned by Reviewer TqkH and Reviewer Bg7V). The goal of a metaheuristic is its generalization to many, though not all, different tasks with promising performance [3]. Many existing metaheuristics such as simulated annealing, iterated local search, tabu search, and memetic search have shown cross-problem generalization [3]. What generalizes is their underlying search logic (i.e., our high-level metaheuristic description), not the specific implementation tailored to a single task.
> > >
> > > [1] Generalizable Heuristic Generation Through LLMs with Meta-Optimization, ICLR 2026
> > > [2] EoH-S: Evolution of Heuristic Set using LLMs for Automated Heuristic Design, AAAI 2026
> > > [3] Handbook of Metaheuristics, 2019
> > >
> > > > On your concerns, “is the improvement brought by them truly significant” and “understand the underlying mechanism of these meta-heuristics.”
> > >
> > > We have compared the designed metaheuristic with strong and widely used baselines. For **BBOP**, we compared it to **CMA-ES** (acknowledged by Reviewer bJP3 as the SOTA method for BBOP) and its two variants; our results show notable superiority.
> > >
> > > The reason is that the results are averaged over five instances (**Sphere, Rosenbrock, Rastrigin, Ackley, and Griewank**). CMA-ES performs insufficiently on **Rastrigin** because its assumption of a single, convex optimum is violated by the function’s many local minima, causing premature convergence before reaching the global optimum. In contrast, the metaheuristic designed from our cross-task MTHS uses a **hybrid population–memory–local-search framework**, which makes it more robust to different instance characteristics.
> > >
> > >
> > > | Method | BBOP |
> > > |---|---:|
> > > | GA | 5.72 |
> > > | CMA-ES | 7.39 |
> > > | CMA-ES active | 5.97 |
> > > | CMA-ES bipop | 0.79 |
> > > | **MTHS** | **0.06** |
> > >
> > > | Method | MaxCut |
> > > |---|---:|
> > > | GA | 5387.4 |
> > > | ILS | 5602.9 |
> > > | Tabu | 5605.0 |
> > > | **MTHS** | **5611.8** |
> > >
> > > | Method | MIS |
> > > |---|---:|
> > > | GA | 29.4 |
> > > | ILS | 28.7 |
> > > | Tabu | 27.3 |
> > > | **MTHS** | **29.6** |
> > >
> > > > On your concern, “Or does it incur substantial overhead at the low-level?”
> > >
> > > When generalizing to other tasks, we only provide the LLM with the metaheuristic designed by MTHS (Appendix E7 shows the prompt; Appendix E9 shows the metaheuristic), and directly sample new implementations for the target tasks **without any low-level search**, as discussed in Section 3.5.
> > >
> > > Regarding the running budget of the metaheuristic itself, we use exactly the same running time (**100s**) for all metaheuristics. Therefore, the generalization cost is reasonable and more efficient than using existing methods.
> > >
> > > We appreciate your valuable feedback and will incorporate all these points into the final version.

---

### Official Review · Reviewer_bJP3 · 2026-03-10

**Soundness:** 3
**Presentation:** 3
**Significance:** 3
**Originality:** 2
**Overall Recommendation:** 4
**Confidence:** 4

**Summary:**

This paper proposes a metaheuristic search framework, which uses a bi-level evolutionary process to search for a task-agnostic metaheuristic that could successfully guide the LLM to generate better performing task-specific program implementations. A knowledge transfer mechanism is designed to enhance the searching across multiple tasks. Experimental results show that the evolved metaheuristics exhibit strong generalization to related tasks.

**Compliance With Llm Reviewing Policy:**

Affirmed.

**Final Justification:**

The response and additioanl experimental results resolve my question, so I increase my score.

**Key Questions For Authors:**

1. In the low-level program search, a program's key function is identified and refined using LLMs. Why does the framework need to identify and refine a "key function" instead of searching the entire program? How does the LLM identify this function, and how can we ensure that no key function is missed?

2. How does the number of tasks involved in the search affect the performance of the discovered metaheuristic? Does including more tasks improve performance, or does it have diminishing returns?

3. It is reasonable that a metaheuristic discovered from a set of combinatorial tasks generalizes well to other combinatorial tasks. However, how can it perform well on BBOP, which have fundamentally different structures and characteristics?

**Limitations:**

yes

**Strengths And Weaknesses:**

Strengths:
1. Separating task-specific programs and task-agnostic metaheuristics during LLM-based heuristic design enables reuse of knowledge and experience gained from previous tasks, enhancing both performance and generalization ability.

2. The discovered metaheuristic outperforms the baselines in generalization across unseen tasks, validating the generalization ability of the proposed framework.

Weaknesses:
1. The bi-level iterative evolutionary process requires a large number of program evaluations and LLM conversations, which can consume significant time and computational resources. In the average running time comparison (Table 7), the stopping criterion for other baselines is maximum iterations or generations, whereas for MTHS it is a time limit. This difference makes the comparison unfair and does not truly demonstrate the time efficiency of MTHS relative to the baselines.

2. The experimental section does not include advanced optimization algorithms such as the Genetic Algorithm (GA) [1] for combinatorial optimization or the Covariance Matrix Adaptation Evolution Strategy (CMA-ES) [2] for black-box optimization. For AHD, Genetic Programming (GP)-based methods [3] and Reinforcement Learning-based methods such as GSF [4] are also not included.

3. The discovered metaheuristic shown on page 28 appears trivial; it is merely a common description of an optimization program and provides little informative insight. Such a metaheuristic may already be implicitly embedded in some general LLMs.

[1] Holland, John H. "Genetic algorithms." Scientific american 267.1 (1992): 66-73.

[2] Auger, Anne, and Nikolaus Hansen. "Tutorial CMA-ES: evolution strategies and covariance matrix adaptation." Proceedings of the 14th annual conference companion on Genetic and evolutionary computation. 2012.

[3] Rivers, Rebecca, and Daniel R. Tauritz. "Evolving black-box search algorithms employing genetic programming." Proceedings of the 15th annual conference companion on Genetic and evolutionary computation. 2013.

[4] Yi, Wenjie, et al. "Automated design of metaheuristics using reinforcement learning within a novel general search framework." IEEE Transactions on Evolutionary Computation 27.4 (2022): 1072-1084.

---

> ### Author Rebuttal · Authors · 2026-03-31
>
> Thank you very much for your time and effort in reviewing our work. We address your concerns point-by-point as follows:
>
> > **W1:** Time and computational cost and fair comparison.
>
> We respond to your concerns in two aspects:
>
> + **MTHS uses fewer evaluations or tokens than existing LLM-driven AHD methods**.  **Appendix E.4** supports exactly this claim: MTHS reduces the number of code evaluations from 1,000 per task to roughly 333 per task on average, and slightly lowers token consumption to about 2.7M versus 3.0M/3.2M/3.1M for EoH/ReEvo/MCTS-AHD on TSP and 2.7M versus 3.2M/3.3M/3.5M on CVRP. In wall-clock time, MTHS is comparable to EoH/ReEvo (about 8h) and much lower than MCTS-AHD (about 40h).
>
> + We have reevaluated the results to use the same running time as the stopping criterion in **Table 7**. A summary is provided in the response to W2. These results clearly demonstrate the superiority of the metaheuristic discovered by MTHS under the same running time (100 seconds).
>
> > **W2:** More comparison with GA, CMA-ES, GP and RL methods.
>
> Thank you. We focus on AHD using LLMs. Our baselines are representative metaheuristics and the most relevant recent LLM-driven AHD methods.
>
> In order to address your concerns, we provide additional results with GA and CMA-ES. Where *activate* and *bipop* represent the CMA-ES variants with activate search and bi-population search for more robust performance. The results show the superiority of MTHS.
>
> |  | CVRP | TSP |
> | --- | --- | --- |
> | OR-Tools SA | 0.72 | 4.31 |
> | OR-Tools TS | 0.34 | 3.64 |
> | OR-Tools GLS | 0.36 | 2.88 |
> | GA | 0.84 | 17.89 |
> | MS | 2.52 | 3.18 |
> | ALNS | 0.82 | 3.75 |
> | TS | 0.54 | 6.89 |
> | MTHS | 0.26 | 0.70 |
>
> |  | BBOB |
> | --- | --- |
> | GA | 5.72 |
> | CMA-ES | 7.39 |
> | CMA-ES active | 5.97 |
> | CMA-ES bipop | 0.79 |
> | MTHS | 0.06 |
>
> We didn't include additional GP and RL experiments, as baseline papers like EoH and ReEvo already provide relevant comparisons (e.g., POMO and LEHD).
>
> We will cite these papers and discuss the differences and advantages in the revised version.
>
> > **W3:** The discovered metaheuristic shown on page 28 appears trivial
>
> We understand the concern, but we would emphasize that it is a **task-agnostic scaffold**, not a full solver. A concise high-level metaheuristic will naturally look more abstract than the instantiated programs. The relevant question is therefore not whether every sentence is individually novel, but whether the scaffold has measurable downstream value. The answer is yes: using the MTHS-designed metaheuristic improves GPT-5-mini on BBOP from 2.86 to 0.32 average score and improves Gemini-2.5-pro on BPP from 1.32% to 0.34% on $n=500$ (**Tables 3-4**). We will revise the presentation to make this distinction explicit.
>
> > **Q1:** Why evolve "key function"?
>
> Searching the entire program at every low-level step would drastically increase the mutation space and make credit assignment much harder. In many heuristic implementations, performance is dominated by one or a few components such as move evaluation, insertion scoring, repair, or neighborhood selection. The design is a pragmatic compromise between tractability and effectiveness, and the ablation in **Table 17** shows that removing low-level refinement worsens TSP from 0.00564 to 0.00636 and CVRP from 0.04122 to 0.04495. We will clarify this point directly in the paper: key-function refinement is a practical design, not a claim of exhaustive program optimization.
>
> > **Q2:** The number of tasks in training.
>
> We expect a trade-off. Adding related tasks can improve the learned metaheuristic by providing a stronger signal for what is genuinely task-agnostic, but adding too many or too diverse tasks will eventually create diminishing returns and increase search cost.
>
> We have conducted additional results trained on four different problems (TSP, CVRP, FSSP, and BBOP). While the results on TSP and CVRP are slightly sacrificed, the results on BBOP are significantly improved, which means that MTHS can still design a general metaheuristic when more problems are used and provide better trade-offs on all training problems.
>
> |  | TSP | CVRP | FSSP | BBOP |
> | --- | --- | --- | --- | --- |
> | MTHS on 3 problems | 0.00564 | 0.04122 | 0.14179 | 0.02880 (test) |
> | MTHS on 4 problems | 0.00994 | 0.04487 | 0.14147 | 0.00003 |
>
> > **Q3:** How can it perform well on BBOP?
>
> Our interpretation is that what transfers to BBOP is not the combinatorial representation itself, but the higher-level search logic: iterative candidate generation, adaptive balance between exploration and exploitation, leveraging historical search experience, and progressive solution refinement. Effective BBOP algorithms also rely on iterative search; many state-of-the-art BBOP methods, for instance, are based on evolutionary or hybrid algorithms. Although the task-specific structures differ, these generic optimization principles are domain-agnostic and thus remain valuable for designing effective on BBOP.

---

> > ### Author Rebuttal · Reviewer_bJP3 · 2026-04-04
> >
> > The response and experimental results resolve my questions.

---

> > > ### Author Response · Authors · 2026-04-04
> > >
> > > Thank you again for your feedback and for confirming that all of your concerns have been fully resolved. We sincerely appreciate your constructive feedback, which helped us strengthen the paper.
> > >
> > > Since **you noted that the rebuttal addressed all of your questions** and that you do not see any major remaining issues with the manuscript, we would like to **respectfully ask whether you might consider updating your evaluation and score to reflect this assessment**, if you feel it is appropriate.
> > >
> > > We believe the additional analyses and evidence provided in the rebuttal further strengthen the paper’s case: they clarify the structure of the discovered metaheuristics, explain when transfer succeeds or fails, confirm budget fairness, and demonstrate the stability of the search process. Together, these additions reinforce the paper’s contribution and empirical strength.
> > >
> > > Of course, we fully respect that this decision is entirely at your discretion. **We simply wanted to kindly ask whether, in light of the fully resolved concerns and clarified weaknesses, the current score (Weak reject: A paper with clear merits, but also some weaknesses, which overall outweigh the merits) still best reflects your updated view of the work**.
> > >
> > > Thank you again for your time, support, and thoughtful consideration.

---

### Official Review · Reviewer_TqkH · 2026-03-12

**Soundness:** 3
**Presentation:** 4
**Significance:** 3
**Originality:** 4
**Overall Recommendation:** 5
**Confidence:** 4

**Summary:**

The paper proposes multi-task hierarchical search (MTHS), an evolutionary framework designed to overcome the task-specificity of current LLM-driven automated heuristic design (AHD). Unlike prior methods that generate monolithic programs, it introduces a hierarchical representation that separates general-purpose metaheuristics from task-specific program implementations. The system employs a two-level evolution process and a knowledge transfer mechanism to co-design these components across multiple optimization tasks simultaneously. Evaluations on four combinatorial optimization tasks and two additional tasks demonstrate that MTHS effectively designs metaheuristics and exhibits strong cross-task generalization.

**Compliance With Llm Reviewing Policy:**

Affirmed.

**Final Justification:**

All my concerns have been addressed. Thus, I maintain my recommendation to accept.

**Key Questions For Authors:**

Discuss the diversity of the evolved metaheuristics. Do they converge to well-known patterns (like Tabu Search), or do the LLMs discover genuinely novel high-level strategies?

The transfer works well for some tasks but fails for others. What properties of tasks influence successful transfer? Could you provide more discussions?

Some LLM-based baselines rely on heuristics reported in previous work rather than re-running the full search. Could the authors clarify whether all methods were evaluated under the same computational budget?

The results focus on best-found solutions. Could the authors report variance or distributional statistics across multiple independent runs to assess the stability of the hierarchical search process?

Is the transfer performed across individuals or within the same individual only? Is transfer always beneficial, or can it hurt performance?

**Limitations:**

yes

**Strengths And Weaknesses:**

Strengths:

The paper tackles what I view as an important limitation of current LLM-driven AHD methods, i.e., their heavy reliance on task-specific program representations, which makes it difficult to reuse knowledge across tasks.

The two-level search (metaheuristic evolution + task-specific program optimization) combined with cross-task knowledge transfer is a coherent design.

The empirical study covers several algorithm design tasks and includes experiments on cross-task generalization. I find the demonstrations where a discovered metaheuristic guides LLMs to design solvers for new tasks particularly interesting, as they suggest the potential for reusable algorithmic knowledge.


Weaknesses:

Although the method claims to produce general metaheuristic structures, the paper provides little analysis of the actual algorithms discovered. Examining their structure, similarities to known metaheuristics, or emergent design patterns would strengthen the contribution.

---

> ### Author Rebuttal · Authors · 2026-03-31
>
> Thank you very much for your time and effort in reviewing our work. We address your concerns point-by-point as follows:
>
> > **W1:** Analysis the designed algorithms and similarities to known metaheuristics.
>
> We agree that the discovered metaheuristics deserve more analysis. In the current submission, the example in Appendix E.9 already shows that MTHS does not merely output a vague description; the discovered ACSS metaheuristic has several concrete components: **(i)** diverse constructive seeding, **(ii)** a cooperative memory of useful substructures, **(iii)** alternating intensification and diversification, **(iv)** explicit feasibility repair, **(v)** adaptive operator selection, and **(vi)** a final polishing stage. We will strengthen the paper with a dedicated analysis section that maps these components to classical metaheuristics such as tabu/memory-based search, ALNS-style destroy-repair, path-relinking/recombination, and iterated local search, while clarifying that the discovered metaheuristic is a reusable hybrid scaffold rather than a re-labeling of any single known method.
>
> > **Q1:** Discuss the diversity of the evolved metaheuristics.
>
> The evidence supports a hybrid outcome rather than convergence to one known pattern. The discovered metaheuristics reuse established search motifs, but MTHS recombines them into higher-level structures that are more general than a standard single-task heuristic. For example, ACSS combines constructive initialization, shared memory, adaptive perturbation, repair, and recombination within one reusable scaffold. The strongest evidence is behavioral: if the result were only a generic pattern, it would be unlikely to outperform existing metaheuristics and specialized LLM-AHD baselines consistently across multiple domains. Yet in Table 1 MTHS is best on 8 of 11 TSP/CVRP benchmark groups, and in Table 2 it achieves the best average FSSP gap (0.24 vs. 0.68 for ILS).
>
> > **Q2:** What properties of tasks influence successful transfer? Could you provide more discussions?
>
> The current results already point to a concrete pattern: transfer is most effective when tasks share *representation-level and operator-level regularities*. TSP, CVRP, FSSP, and BPP all admit iterative construction/improvement, reuse of partial structures, and alternating intensification/diversification. This is why the discovered metaheuristic transfers well to BPP and BBOP (**Table 4**).
>
> In contrast, ASP is structurally less aligned because no iterative search is used (**Table 5**). We will revise the paper to state this conclusion directly: transfer depends on shared decomposition, neighborhood structure, and feasibility-repair patterns, and it is *not* uniformly beneficial across all target domains.
>
> > **Q3:** Clarify whether all methods were evaluated under the same computational budget.
>
> Thank you. We will clarify this more carefully. For MTHS, STHS, and the re-run LLM baselines (EoH, ReEvo, MCTS-AHD with GLS), we used the same search budget of 1,000 program evaluations with the same LLM GPT-5-mini. For ACO-based baselines, we adopted the best heuristics reported in prior work instead of re-running the full search; we did this because those baselines were already available from the prior study and because GLS was shown there to be stronger than ACO.
>
> > **Q4:** Report variance or distributional statistics across multiple independent runs to assess the stability of the hierarchical search process.
>
> We follow the common protocol in prior LLM-driven AHD work and report the best discovered heuristic from a fixed search budget. We have conducted three independent runs; the results (optimality gaps) are as follows:
>
> | Run | TSP | CVRP | FSSP |
> | --- | --- | --- | --- |
> | 1 | 0.0056 | 0.0412 | 0.1417 |
> | 2 | 0.0059 | 0.0395 | 0.1390 |
> | 3 | 0.0030 | 0.0429 | 0.1368 |
> | Mean | 0.0048 | 0.0412 | 0.1392 |
> | Std. Dev. | 0.0013 | 0.0013 | 0.0020 |
>
> These results show that MTHS achieves consistently low gaps across runs, with small standard deviations relative to the baseline performance levels.
>
> > **Q5:** Is the transfer performed across individuals or within the same individual only? Is transfer always beneficial, or can it hurt performance?
>
> Transfer is *not* restricted to a single program in isolation. In the current formulation, for each source task, we select the best-performing program from the current population, adapt it to other target tasks, and keep the adapted version only if it improves the target-task score. Therefore, transfer can, in principle, be harmful before evaluation, but harmful transfers are filtered out by the accept-if-better rule and do not survive in the population. This is also consistent with the ablation in **Table 17**: removing transfer degrades TSP from 0.00564 to 0.00570 and CVRP from 0.04122 to 0.04446, while the selection rule mitigates persistent negative transfer.

---

> > ### Author Rebuttal · Reviewer_TqkH · 2026-04-02
> >
> > I thank the authors for their response. All my concerns have been addressed. I have also read the other reviewers' comments and the authors' replies, and I do not see any major remaining issues with this manuscript. Thus, I maintain my recommendation to accept.

---

> > > ### Author Response · Authors · 2026-04-03
> > >
> > > Thank you very much for your thorough and constructive review. We truly appreciate your insightful comments and questions, which have helped us strengthen the paper.
> > >
> > > We are pleased that our rebuttal addressed your concerns regarding the analysis of discovered metaheuristics, transfer properties, computational budgets, variance statistics, and the transfer mechanism. We will incorporate the additional analyses (e.g., diversity of evolved heuristics, stability across runs, and conditions for successful transfer) into the revised version of the paper.

---

### Decision · Program_Chairs · 2026-04-30

**Decision:**

Accept (regular)

**Comment:**

The reviewers appreciate that this paper addresses the important challenge of discovering generalizable heuristics in LLM-driven Automated Heuristic Design (AHD). Reviewers consider the novel two-level representation a strength to learn simutaneously a task-agnostic metaheuristic and task-specific programs. In the initial review, reviewers raised several shared concerns, including:
- Interpretability of the generated metaheuristics
- Computation cost and fair comparison with baselines
- A lack of analysis on when positive versus negative transfer occurs
- Abscence of comparison with more advanced optimization algorithms
- Insufficient analysis of generalization across varying problem scales and as the number of tasks increases

In their rebuttal, the authors provided additional experiments involving new problems and stronger baselines. These additions successfully addressed the majority of concerns, prompting multiple reviewers to raise their scores. A few minor concerns remained only partially addressed:
- Generalization analysis across problem sizes: authors clarified that this submission was focused on cross-problem generalization, acknowledging the challenge of out-of-distribution generalization across problem scales
- Performance scaling with the number of tasks: while one reviewer agreed the rebuttal demonstrated the "logical consistency of the MTHS framework in multi-task scenarios," they maintained that further empirical validation was expected.

Following the rebuttal, one reviewer introduced a new concern regarding incremental contribution against prior works (EoH and ReEvo) and include additional criticism:
- Imbalance in baseline comparisons: lack of comparison with EoH and ReEvo.
- Lack of new empirical evidence in rebuttal: specifically regarding "negative transfer" and "multi-task scalability"
- Generalization Boundaries: failure to generalize to the Admissible Set Problem (ASP)

However, the AC notes that these late-stage concerns are not well supported by the initial reviews or the broader consensus reached during discussion with the other reviewers. The proposed two-level representation has been widely acknowledged by the review panel as the a distinct contribution against over recent baselines such as EoH and ReEvo. Morever, these methods are explicitly included as primary baselines in the experiments, alongside classical optimization algorithms. Although the authors did not have the opportunity to respond to the late review update, the analysis of negative transfer and mixed generalization results on ASP were already discussed during the rebuttal phase and were deemed resolved by other reviewers.

Based on the comprehensive discussion during the rebuttal, the AC consider the majority of the initial concerns have been successfully resolved. The remaining weaknesses - namely, the limited generalization across problem scales, instances of mixed/negative transfer, and a lack of experiments scaling to a larger number of tasks - should be explicitly discussed in the final revision, but they do not constitute grounds for rejection.